# POISONING ATTACKS WITH GENERATIVE ADVERSARIAL NETS

## ABSTRACT

Machine learning algorithms are vulnerable to poisoning attacks: An adversary can inject malicious points in the training dataset to influence the learning process and degrade the algorithm's performance. Optimal poisoning attacks have already been proposed to evaluate worst-case scenarios, modelling attacks as a bi-level optimization problem. Solving these problems is computationally demanding and has limited applicability for some models such as deep networks. In this paper we introduce a novel generative model to craft systematic poisoning attacks against machine learning classifiers generating adversarial training examples, i.e. samples that look like genuine data points but that degrade the classifier's accuracy when used for training. We propose a Generative Adversarial Net with three components: generator, discriminator, and the target classifier. This approach allows us to model naturally the detectability constrains that can be expected in realistic attacks and to identify the regions of the underlying data distribution that can be more vulnerable to data poisoning. Our experimental evaluation shows the effectiveness of our attack to compromise deep networks, bypassing different type of existing defences against data poisoning.

## 1 INTRODUCTION

Despite the advancements and the benefits of machine learning, it has been shown that learning algorithms are vulnerable and can be the target of attackers, who can gain a significant advantage by exploiting these vulnerabilities (Huang et al., 2011). At training time, learning algorithms are vulnerable to poisoning attacks, where small fractions of malicious points injected in the training set can subvert the learning process and degrade the performance of the system in an indiscriminate or targeted way. Data poisoning is one of the most relevant and emerging security threats in applications that rely upon the collection of large amounts of data in the wild (Joseph et al., 2013). Some applications rely on the data from users' feedback or untrusted sources of information that often collude towards the same malicious goal. For example, in IoT environments sensors can be compromised and adversaries can craft coordinated attacks manipulating the measurements of neighbour sensors evading detection (Illiano et al., 2016). In many applications *curation* of the whole training dataset is not possible, exposing machine learning systems to poisoning attacks.

In the research literature optimal poisoning attack strategies have been proposed against different machine learning algorithms (Biggio et al., 2012; Mei & Zhu, 2015; Muñoz-González et al., 2017; Jagielski et al., 2018), allowing to assess their performance in worst-case scenarios. These attacks can be modelled as a bi-level optimization problem, where the outer objective represents the attacker's goal and the inner objective corresponds to the training of the learning algorithm with the poisoned dataset. Solving these bi-level optimization problems is challenging and can be computationally demanding, especially for generating poisoning points at scale. This limits its applicability against some learning algorithms such as deep networks or where the training set is large. In many cases, if no detectability constraints are considered, the poisoning points generated are outliers that can be removed with data filtering (Paudice et al., 2018a). Furthermore, such attacks are not realistic as real attackers would aim to remain undetected in order to be able to continue subverting the system in the future. As shown in (Koh et al., 2018), detectability constraints for these optimal attack strategies can be modelled, however they further increase the complexity of the attack, limiting even more the application of these techniques.

Taking an entirely different and novel approach, in this paper we propose a poisoning attack strategy against machine learning classifiers with Generative Adversarial Nets (GANs) (Goodfellow et al., 2014). This allows us to craft poisoning points in a more systematic way, looking for regions of the data distribution where the poisoning points are more influential and, at the same time, difficult to detect. Our proposed scheme, *pGAN*, consists on three components: generator, discriminator and target classifier. The generator aims to generate poisoning points that maximize the error of the target classifier but minimize the discriminator's ability to distinguish them from genuine data points. The classifier aims to minimize some loss function evaluated on a training dataset that contains a fraction of poisoning points. As in a standard GAN, the problem can be formulated as a minimax game. pGAN allows to systematically generate *adversarial training examples* (Koh & Liang, 2017), which are similar to genuine data points but that can degrade the performance of the system when used for training. The use of a generative model allows us to produce poisoning points at scale, enabling poisoning attacks against learning algorithms where the number of training points is large or in situations where optimal attack strategies with bi-level optimization are intractable or difficult to compute, as it can be the case for deep networks. Additionally, our proposed model also includes a mechanism to control the detectability of the generated poisoning points. For this, the generator maximizes a convex combination of the losses for the discriminator and the classifier evaluated on the poisoning data points. Our model allows to control the aggressiveness of the attack through a parameter that controls the weighted sum of the two losses. This induces a trade-off between effectiveness and detectability of the attack. In this way, pGAN can be applied for systematic testing of machine learning classifiers at different risk levels. Our experimental evaluation in synthetic and real datasets shows that pGAN is capable of compromising different machine learning classifiers bypassing different defence mechanisms, including outlier detection Paudice et al. (2018a), Sever Diakonikolas et al. (2019), PCA-based defences Rubinstein et al. (2009) and label sanitization Paudice et al. (2018b). We analyse the trade-off between detectability and effectiveness of the attack: Too conservative strategies will have a reduced impact on the target classifier but, if the attack is too aggressive, most poisoning points can be detected as outliers.

## 2 RELATED WORK

The first practical poisoning attacks were proposed in the context of spam filtering and anomaly detection (Nelson et al., 2008; Kloft & Laskov, 2012). But these attacks do not easily generalize to different learning algorithms. Biggio et al. (2012) presented a more systematic approach, modelling optimal poisoning attacks against SVMs for binary classification as a bi-level optimization problem, which can be solved by exploiting the Karush-Kuhn-Tucker conditions in the inner problem. A similar approach is proposed by Xiao et al. (2015) for poisoning embedded feature selection methods, including LASSO, ridge regression, and elastic net. Mei & Zhu (2015) proposed a more general framework to model and solve optimal poisoning attacks for convex classifiers. They exploit the implicit function theorem to compute the gradients required to solve the corresponding bi-level optimization problem. Muñoz-González et al. (2017) proposed back-gradient optimization to estimate the gradients required to solve bi-level optimization problems for optimal poisoning attacks against multi-class classifiers. This approach allows to attack a broader range of learning algorithms and reduces the computational complexity with respect to previous works. However, all these techniques are limited to compromise deep networks trained with a large number of training points, where many poisoning points are required even to compromise a small fraction of the training dataset. Previous attacks did not model explicitly appropriate detectability constraints. Thus, the resulting poisoning points can be far from the genuine data distribution and can be easily identified as outliers (Paudice et al., 2018a; Steinhardt et al., 2017; Paudice et al., 2018b). Recently, Koh et al. (2018) showed that it is still possible to craft attacks capable of bypassing outlier-detection-based defences with an iterative constrained bi-level optimization problem, where, at each iteration, the constraints change according to the current solution of the bi-level problem. However, the high computational complexity of this attack limits its practical application in many scenarios.

Koh & Liang (2017) proposed a different approach to craft targeted attacks against deep networks by exploiting influence functions. This approach allows to create adversarial training examples by learning small perturbations that, when added to some specific genuine training points, change the predictions for a target set of test points. Shafahi et al. (2018) showed that it is possible to perform targeted attacks when the adversary is not in control of the labels for the poisoning points. Yang et al. (2017) introduced a poisoning attack with generative models using autoencoders to generate the

malicious points. Although this method is more scalable than attacks based on bi-level optimization, the authors do not provide a mechanism to control the detectability of the poisoning points.

# 3 POISONING ATTACKS WITH GENERATIVE ADVERSARIAL NETS

Our model, *pGAN*, is a GAN-based model with three components (generator, discriminator and target classifier) to generate systematically adversarial training examples. First, we shortly describe the considered model for the attacker. Then, we introduce the formulation of pGAN and, finally, we provide some practical considerations for the implementation of pGAN.

## 3.1 ATTACKER'S MODEL

The *attacker's knowledge* of the targeted system depends on different aspects: the learning algorithm, the objective function optimized, the feature set or the training data. In our case we consider *perfect knowledge* attacks, where we assume the attacker knows everything about the target system: the training data, the feature set, the loss function and the machine learning model used by the victim. Although unrealistic in most practical scenarios, this assumption allows us to perform worst-case analysis of the performance of the system under attack. However, our proposed attack strategy also supports *limited knowledge*, exploiting the transferability property of poisoning attacks (Muñoz-González et al., 2017). For the *attacker's capabilities*, we consider here a *causative attack* (Barreno et al., 2006; 2010), where the attacker can manipulate a fraction of the training data to influence the learning algorithm. We assume that the attacker can manipulate all the features to craft the poisoning points as long as the resulting points are within the feasible domain for the distribution of genuine training points. Finally, we also assume that the attacker can also control the labels of the injected poisoning points.

## 3.2 PGAN

In a multi-class classification task, let $\mathcal{X} \in \mathcal{R}^d$ be the $d$-dimensional feature space, where data points $\mathbf{x}$ are drawn from a distribution $p_x(\mathbf{x})$ and $\mathcal{Y}$ is the space of class labels. The learning algorithm, $\mathcal{C}$, aims to learn the mapping $f : \mathcal{X} \to \mathcal{Y}$ by minimizing a loss function, $\mathcal{L}_{\mathcal{C}}$, evaluated on a set of training points $\mathcal{S}_{tr}$. The objective of the attacker is to introduce a fraction, $\lambda \in (0, 1)$, of malicious points in $\mathcal{S}_{tr}$ to maximize $\mathcal{L}_{\mathcal{C}}$ when evaluated on the poisoned training set.

The Generator, $\mathcal{G}$, aims to generate poisoning points by learning a data distribution that is effective at increasing the error of the target classifier, but that is also close to the distribution of genuine data points, i.e. the generated poisoning points are similar to honest data points to evade detection. Thus, $\mathcal{G}$ receives some noise $\mathbf{z} \sim p_z(\mathbf{z}|\mathbf{Y}_p)$ as input and implicitly defines a distribution of poisoning points, $p_p(\mathbf{x})$, which is the distribution of the samples $\mathcal{G}(\mathbf{z}|\mathbf{Y}_p)$ conditioned on $\mathbf{Y}_p \subset \mathcal{Y}$, the set of target class labels for the attacker. The Discriminator, $\mathcal{D}$, aims to distinguish between honest training data and the generated poisoning points. It estimates $\mathcal{D}(\mathbf{x}|\mathbf{Y}_p)$, the probability that $\mathbf{x}$ came from the genuine data distribution $p_x$ rather than $p_p$. As in $\mathcal{G}$, the samples used in the discriminator are conditioned on the set of labels $\mathbf{Y}_p$. The Classifier, $\mathcal{C}$, is representative for the attacked algorithm. In perfect knowledge attacks $\mathcal{C}$ can have the same structure as the actual target classifier. For black-box attacks we can exploit attack transferability, and then, use $\mathcal{C}$ as a surrogate model that can be somewhat similar to the actual (unknown) classifier. During the training of pGAN, $\mathcal{C}$ is fed honest and poisoning training points from $p_x$ and $p_p$ respectively, where the fraction of poisoning points is controlled by a parameter $\lambda \in [0, 1]$.

In contrast to traditional GAN schemes, $\mathcal{G}$ in pGAN plays a game against both $\mathcal{D}$ and $\mathcal{C}$. This can also be formalized as a minimax game where the maximization problem involves both $\mathcal{D}$ and $\mathcal{C}$. Similar to conditional GANs (Mirza & Osindero, 2014), the objective function for $\mathcal{D}$ (which also depends on $\mathcal{G}$) can be written as:

$$\mathbb{V}(\mathcal{D}, \mathcal{G}) = \mathbb{E}_{\mathbf{z}\sim p_z(\mathbf{z}|\mathbf{Y}_p)}[\log(1 - \mathcal{D}(\mathcal{G}(\mathbf{z}|\mathbf{Y}_p)))] + \mathbb{E}_{\mathbf{x}\sim p_x(\mathbf{x}|\mathbf{Y}_p)}[\log(\mathcal{D}(\mathbf{x}|\mathbf{Y}_p)]. \tag{1}$$

The objective function for $\mathcal{C}$ is given by:

$$\mathbb{W}(\mathcal{C}, \mathcal{G}) = -\left(\lambda \, \mathbb{E}_{\mathbf{z}\sim p_z(\mathbf{z}|\mathbf{Y}_p)}[\mathcal{L}_{\mathcal{C}}(\mathcal{G}(\mathbf{z}|\mathbf{Y}_p))] + (1 - \lambda) \, \mathbb{E}_{\mathbf{x}\sim p_x(\mathbf{x})}[\mathcal{L}_{\mathcal{C}}(\mathbf{x})]\right), \tag{2}$$

where $\lambda$ is the fraction of poisoning points introduced in the training dataset and $\mathcal{L}_\mathcal{C}$ is the loss function used to train $\mathcal{C}$. Note that the poisoning points in (2) belong to a subset of poisoning class labels $\mathbf{Y}_p$, whereas the genuine points used to train the classifier are from all the classes. The objective in (2) is just the negative loss used to train $\mathcal{C}$ evaluated on a mixture of honest and poisoning points (from the set of classes in $\mathbf{Y}_p$) controlled by $\lambda$.

Given (1) and (2), pGAN can then be formulated as the following minimax problem:

$$\min_\mathcal{G} \max_{\mathcal{D},\mathcal{C}} \ \alpha \, \mathbb{V}(\mathcal{D},\mathcal{G}) + (1-\alpha) \, \mathbb{W}(\mathcal{C},\mathcal{G}) \qquad (3)$$

with $\alpha \in [0,1]$. In this case, the maximization problem can be seen as a multi-objective optimization problem to learn the parameters of both the classifier and the discriminator. Whereas for $\mathcal{C}$ and $\mathcal{D}$ the objectives are decoupled, the generator optimizes a convex combination of the two objectives in (1) and (2). The parameter $\alpha$ controls the importance of each of the two objective functions towards the global goal. So, for high values of $\alpha$, the attack points will prioritize evading detection, rendering attacks with (possibly) a reduced effectiveness. Note that for $\alpha = 1$ we have the same minimax game as in a standard conditional GAN (Mirza & Osindero, 2014). On the other hand, low values of $\alpha$ will result in attacks with higher impact in the classifier's performance. However the generated poisoning points will be more detectable by outlier detection systems. For $\alpha = 0$, pGAN does not consider any detectability constraint and the generated poisoning points are only constrained by the output activation functions in the $\mathcal{G}$. In this case pGAN can serve as a suboptimal approximation of the optimal attack strategies in (Biggio et al., 2012; Mei & Zhu, 2015; Muñoz-González et al., 2017) where no detectability constraints are imposed.

Similar to (Goodfellow et al., 2014) we train pGAN following a coordinated gradient-based strategy to solve the minimax problem in (3). We sequentially update the parameters of the three components using mini-batch stochastic gradient descent/ascent. For the generator and the discriminator data points are sampled from the conditional distribution on the subset of poisoning labels $\mathbf{Y}_p$. For the classifier, honest data points are sampled from the data distribution including all the classes. A different number of iterations can be considered for updating the parameters of the three blocks. The details of the training algorithm are provided in Appendix A.

### 3.3 PRACTICAL CONSIDERATIONS

The formulation of pGAN in (3) allows to perform both error-generic and error-specific poisoning attacks (Muñoz-González et al., 2017), which aim to increase the error of the classifier in an indiscriminate or a specific way. However, the nature of these errors can be limited by $\mathbf{Y}_p$, i.e. the classes for which the attacker can inject poisoning points. To generate targeted attacks or to produce more specific types of errors in the system we need to use a surrogate model for the target classifier in pGAN, including only the classes or samples considered in the attacker's goal. For example, if the attacker wants to inject poisoning points labelled as $i$ to increase the classification error for class $j$, we can use a binary classifier in pGAN considering only classes $i$ and $j$, where the generator aims to produce samples from class $i$. As in other GAN schemes, pGAN can also be difficult to train and can be prone to mode collapse. To mitigate these problems, we used in our experiments some of the standard techniques proposed to improve GANs training, such as dropout or batch-normalization (Salimans et al., 2016). We also applied one-side label smoothing, not only for the labels in the discriminator but also for the labels of the genuine points in the classifier. As suggested by Goodfellow et al. (2015), to avoid small gradients for $\mathcal{G}$ from the discriminator's loss function (1), especially in early stages where the quality of the samples produced by $\mathcal{G}$ is poor, we train $\mathcal{G}$ to maximize $\log(\mathcal{D}(\mathcal{G}(\mathbf{z}|\mathbf{Y}_p)))$ rather than minimizing $\log(1 - \mathcal{D}(\mathcal{G}(\mathbf{z}|\mathbf{Y}_p)))$.

In contrast to standard GANs, in pGAN the learned distribution of poisoning points $p_p$ is expected to be different from the distribution of genuine points $p_x$. Thus, the accuracy of the discriminator in pGAN will always be greater that $0.5$. Then, the stopping criteria for training pGAN cannot be based on the discriminator's accuracy. We need to find a saddle point where the objectives in (1) and (2) are maximized for $\mathcal{D}$ and $\mathcal{G}$ respectively (i.e. pGAN finds local maxima) and the the combined objective in (3) is minimized w.r.t. $\mathcal{G}$ (i.e. pGAN finds a local minimum). Finally, the value of $\lambda$ plays an important role in the training of pGAN. If $\lambda$ is small, the gradients for $\mathcal{G}$ from the classifier's loss in (2) can be very small compared to the gradients from the discriminator's loss in (1). Thus, the generator focuses more on evading detection by the discriminator rather than increasing the error of the target classifier, resulting in blunt attacks. Then, even if the expected fraction of poisoning

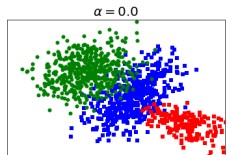 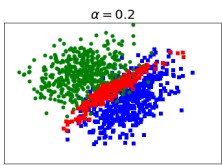 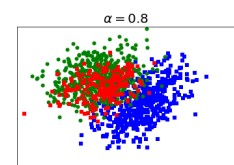 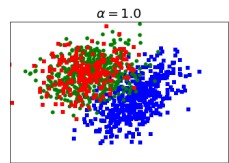

Figure 1: Synthetic experiment: Distribution of genuine (green and blue dots) and poisoning (red dots) data points for different values of $\alpha$. The poisoning points are labelled as green.

points to be injected in the target system is small, larger values of $\lambda$ are preferred to generate more successful poisoning attacks. In our experiments in Sect. 4 we analyse the effectiveness of the attack as a function of $\lambda$.

## 4 EXPERIMENTS

### 4.1 SYNTHETIC EXPERIMENT

To illustrate how pGAN works we first performed a synthetic experiment with a binary classification problem, generating two bivariate Gaussian distributions that slightly overlap. We trained pGAN for different values of $\alpha$ with 500 training points from each Gaussian distribution. We targeted a logistic regression classifier with $\lambda = 0.8$. In Fig. 1 we show the distribution of poisoning (red dots) and genuine (green and blue dots) data points. The poisoning points are labelled as the green data points. Thus, $\mathcal{G}$ aims to generate malicious points, similar to the green ones (i.e. $\mathcal{D}$ aims to discriminate between red and green data points). For $\alpha = 1$ we have the same result as in a standard GAN, so that the distribution of red points matches the distribution of the green ones. But, as we decrease the value of $\alpha$, the distribution of red points shifts towards the region where both green and blue distributions overlap. We can observe that for $\alpha = 0.2$ the poisoning points are still close to genuine green points, i.e. we cannot consider the red points as outliers in most cases. For $\alpha = 0$ the generator does not have detectability constraints, focusing only on increasing the error of the classifier. It is interesting to observe that, in this case, pGAN does not produce points interpolating the distribution of the two genuine classes, but the distribution learned by the generator is far from the region where the distributions of the blue and green points overlap.[1] This suggests that for $\alpha \neq 0$ pGAN is not just producing a simple interpolation between the two classes, but $\mathcal{G}$ looks for regions close to the decision boundary where the classifier is weaker. The complete details of the experiment and the effect on the decision boundary after injecting the poisoning points can be found in Appendix B.

### 4.2 ATTACK EFFECTIVENESS

We performed our experimental evaluation on *MNIST* (LeCun et al., 1998), *Fashion-MNIST* (FM-NIST) (Xiao et al., 2017), and CIFAR-10 Krizhevsky (2009) datasets, using Deep Neural Networks (DNNs) for MNIST and FMNIST and Convolutional Neural Networks (CNNs) for CIFAR. All details about the datasets used and the experimental settings in our experiments are described in Appendix C. To test the effectiveness of pGAN to generate *stealthy* poisoning attacks we applied the defence strategy proposed by Paudice et al. (2018a): We assumed that the defender has a fraction of trusted data points that can be used to train one outlier detector for each class in the classification problem. Thus, we pre-filter the (genuine and malicious) training data points with these outlier detectors before training. As in (Paudice et al., 2018a) we used the distance-based anomaly detector proposed by Wu & Jermaine (2006), which was proven to be effective against optimal poisoning attacks (Biggio et al., 2012; Muñoz-González et al., 2017). The *outlierness* score is computed based on the euclidean distance between the tested data point and its $k$-nearest neighbours from a subset of $s$ points, which are sampled without replacement from the set of points used to train the outlier detector. In our experiments we used the same values proposed in (Paudice et al., 2018a): $k = 5$ for the number of neighbours and $s = 20$ for the number of training points to be sampled. We set the threshold of the outlier detector so that the $\alpha$-percentile is 0.95. The $\alpha$-percentile controls the fraction of genuine points that is expected to be retained after applying the outlier detector (i.e. 95%

---

[1]Note that the result would be significantly different if the target classifier were non-linear.

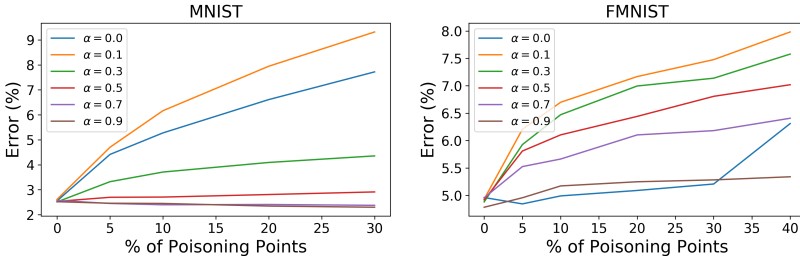

Figure 2: Test classification error (%) as a function of the percentage of poisoning points using pGAN with different values of $\alpha$ for MNIST (left) and FMNIST (right).

in our case). To provide a better understanding of the behaviour of pGAN we first trained and tested our attack targeting binary classifiers. For this, in MNIST we selected digits 3 and 5, for FMNIST we picked the classes *sneaker* and *ankle boot* and, for CIFAR, *automobile* and *truck* classes. The poisoning points were labelled as 5, *ankle boot* and *truck* respectively.

First, we analysed the effectiveness of the attack as a function of $\alpha$. For each dataset we trained 5 different generators for each value of $\alpha$ explored, $[0.0, 0.1, 0.3, 0.5, 0.7, 0.9]$. We set $\lambda = 0.9 \cdot \Pr(Y_p)$, where $\Pr(Y_p)$ is the prior probability of the samples from the poisoning class, $Y_p$ (i.e. digit 5, *ankle boot*, and *truck*). For testing, we used 500 (genuine) samples per class to train the outlier detectors and 500 samples per class to train a separate classifier. We evaluated the effectiveness of the attack varying the fraction of poisoning points, exploring values in the range $0 - 40\%$. To preserve the ratio between classes we substitute genuine samples from the poisoning class with the malicious points generated by pGAN (rather than adding the poisoning points to the given training dataset). For each pGAN generator and for each value of the fraction of poisoning points explored, we did 10 independent runs with independent splits for the outlier detectors and the classifier training sets.

**MNIST and FMNIST:** In Fig. 2 we show the test classification error for MNIST and FMNIST as a function of the fraction of poisoning points averaged over the 5 generators and the 10 runs for each generator. In MNIST, the attack is more effective for $\alpha = 0.1$, increasing the error from $2.5\%$ when there's no attack to more than $12\%$ when $40\%$ of the training dataset is compromised. For bigger values of $\alpha$ the effect of the attack is more limited. For $\alpha = 0$, i.e. when no detectability constraints are considered the effect of the attack is more limited than for $\alpha = 0.1$, although in this case, the points capable of bypassing the outlier detector can still produce some degradation in the target system. Similarly, for FMNIST the attack with $\alpha = 0.1$ produces more effective poisoning data points, although the overall effect of the attack is more limited compared to MNIST. For $\alpha = 0$ the attack is mitigated by the outlier detector in most cases, and has only some effectiveness for larger fractions of poisoning points. It is interesting to observe that, despite the baseline error (i.e. when there is no attack) is lower for MNIST ($2.5\%$ vs $4.75\%$ in FMNIST), it is more difficult to poison FMNIST. This suggests that the impact of the attack not only depends on the separation between the two classes but also on the topology of the classification problem. In Fig. 3 we show some of the poisoning examples generated by pGAN (with $\alpha = 0.3$). For MNIST the malicious data points (labelled as 5) exhibit features from both, digits 3 and 5. In some cases, although the poisoning digits are similar to a 3, it is difficult to automatically detect these points as outliers, as many of the pixels that represent these malicious digits follow a similar pattern compared to genuine 5s, i.e. they just differ in the upper trace of the generated digits. In other cases, the malicious digits look like a 5 that have some characteristics that make them closer to 3s. In the case of FMNIST, the samples generated by pGAN (labelled as *ankle boots*) can be seen as an interpolation of the two classes. The malicious images look like *high-top sneakers* or *low-top ankle boots*. Thus, it is difficult to detect them as malicious points, as they clearly resemble some of the genuine *ankle boots* in the genuine training set. Actually, for some of the genuine images, it is difficult to identify them as a sneaker or an ankle boot. More examples for different values of $\alpha$ are also shown in Appendix D.

**CIFAR:** The results in Fig. 4(left) show that for $\alpha = 0$, i.e. with no detectability constraints, the attack is significantly more effective than for the other values of $\alpha$ explored. In this case, we observed that the outlier detector can be easily bypassed in this dataset, as the number of features and the complexity of the classification tasks are higher compared to MNIST and FMNIST. This

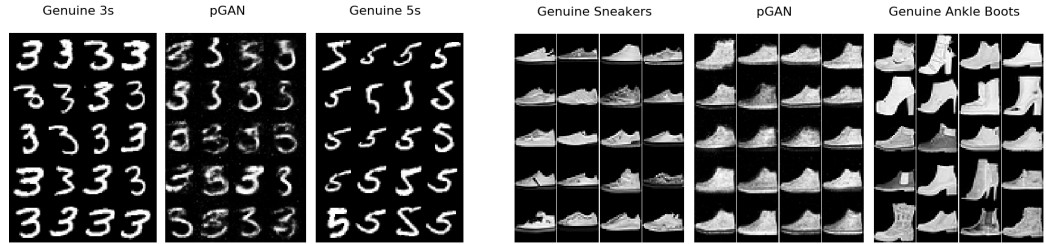

Figure 3: Examples from pGAN (with $\alpha = 0.3$) for MNIST (left) and FMNIST (right).

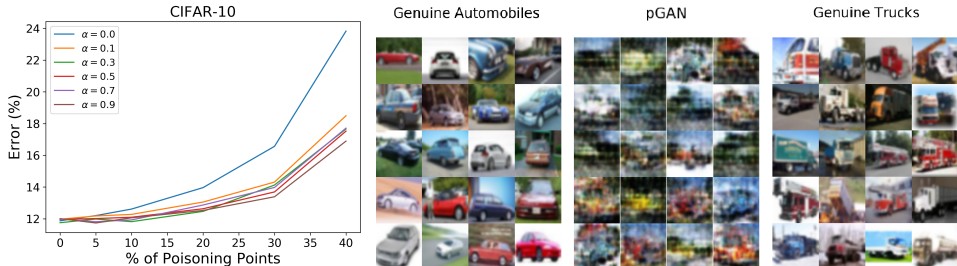

Figure 4: (Left) Average error on CIFAR as a function of $\alpha$. (Right) Examples of images generated by pGAN for CIFAR ($\alpha = 0.3$).

shows some of the limitations of some existing defences to mitigate data poisoning attacks. We can observe that in this case, the test classification error increases from 12% to 24% after injecting 40% of poisoning points. The increase is more moderate for larger values of $\alpha$. In Fig. 4(right) we show some of the examples generated by pGAN with $\alpha = 0.3$, which exhibit characteristics from the two classes: cars and trucks.

**Outlier detection:** In Fig. 5 (centre) we show the fraction of data points pre-filtered by the outlier detectors in MNIST dataset as a function of $\alpha$. We explored two values for the $\alpha$-percentile (the threshold of the detectors): 0.90 and 0.95. As expected, the fraction of rejected genuine data points is, on average, 10% and 5% respectively. However, the fraction of rejected malicious points for $\alpha \geq 0.1$ is smaller than for the genuine points for the two detectors. This is because the generator pays less attention to samples that are in low density regions for the data distribution of the genuine points, and then, the generated poisoning points are *conservative*. For $\alpha = 0$ the fraction of rejected malicious points is also not very high. This can be due to the similarity between the two classes. Then, even if the generated poisoning points, labelled as 5, look like a 3 they are still close to the distribution of genuine 5s when targeting a non-linear classifier.

**Sensitivity w.r.t. $\lambda$:** For analysing the sensitivity of pGAN w.r.t. $\lambda$, the fraction of poisoning points used for $\mathcal{C}$, we performed an experiment on MNIST dataset (digits 3 and 5). We set $\alpha = 0.2$ and explored different values for $\lambda' = \lambda/\mathrm{Pr}(Y_p)$ ranging from 0.1 to 1. With the same experimental settings as before we trained 5 generators for each value of $\lambda'$. We also tested the effectiveness of the attack on a separate classifier, with 10 independent runs for each generator and value of $\lambda'$ explored. For the attacks we injected 20% of poisoning points. In Fig. 5 (left) we show the averaged classification error on the test dataset as a function of $\lambda'$. We can observe that, for small $\lambda'$, the effect of the attack is more limited. The reason is that, when training pGAN the effect of the poisoning points on $\mathcal{C}$ is very reduced, and then, the gradients of (2) w.r.t. the parameters of $\mathcal{G}$ can be very small compared to the gradients coming from the discriminator. Then, $G$ focuses more on optimizing the discriminator's objective. In this case, even for $\lambda' = 1$ the attack is still effective, just slightly decreasing the error rate compared to $\lambda' = 0.9$.

**Comparison with label flipping:** Comparison with existing poisoning attacks in the research literature is challenging: Optimal poisoning attacks as in Muñoz-González et al. (2017) are computationally very expensive for the size of the networks and datasets used in our experiments in Fig. 2. This is even worse if we consider detectability constraints as in Koh et al. (2018). On the other side, comparing with standard label flipping results in an unfair comparison for pGAN, as label flipping

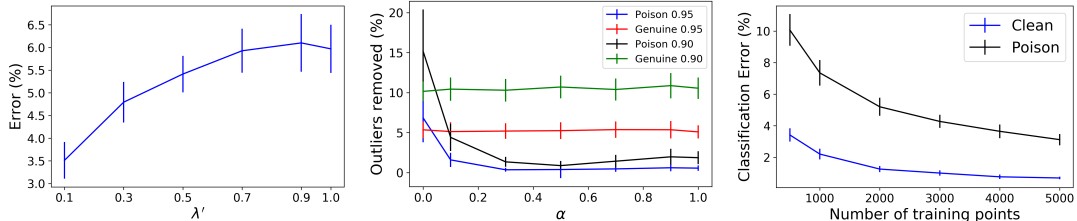

Figure 5: (Left) Average test error on MNIST as a function of $\lambda' = \lambda/\text{Pr}(Y_p)$. (Centre) Outlier detection on MNIST as a function of $\alpha$ for $\alpha$-percentiles of $0.95$ and $0.90$. (Right) Average test error on MNIST as a function of of the number of training examples for a clean and a poisoned classifier (with $20\%$ of poisoning points).

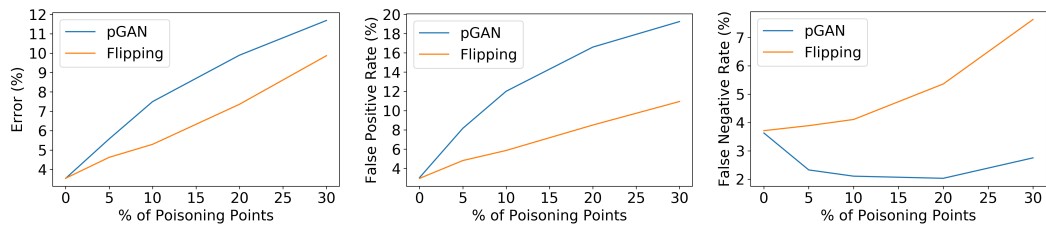

Figure 6: Average test error (left), false positive (centre) and false negative rates (right) as a function of the percentage of poisoning points for pGAN and label flipping attacks on MNIST.

do not consider detectability constraints. In other words, we can expect label flipping to be more effective than pGAN when no defence is applied, but this attack is clearly more detectable (Paudice et al., 2018b). To provide a fairer comparison, we implemented an heuristic for generating label flipping attacks with detectability constraints. Thus, we flipped the labels of training samples from the target class that are closer to the source class. For this, we computed the distance of the training points from the target class to the mean of the training points of the source class. Then, we flipped the labels of the closest points, so that the malicious points should be more difficult to detect. In Fig. 6 we show the comparison of this label flipping strategy with pGAN ($\alpha = 0.1$) for MNIST, using the same settings as in the experiment in Fig. 2. We can observe that pGAN is more effective than the label flipping attack and that the effect of the two attacks is different. Label flipping increases both the false positive and false negative rates of the target classifier, whereas pGAN aims only to increase the false positive rate, i.e. pGAN is producing an error-specific attack, giving the attacker more control on the kind of errors to be produced in the classifier.

**Attack effectiveness as a function of the number of training points:** In Fig. 5 (right) we show how the number of training data points impact the effect of the attack. For this, we trained 5 pGAN generators with $\alpha = 0.1$ and tested on classifiers with different number of training points ranging from $1,000$ to $10,000$ and injecting $20\%$ of poisoning points. For each generator and value of the number of training points explored we did 5 independent runs. We also used 500 samples per class to train the outlier detectors. The results in Fig. 5 (right) show that the difference in performance between the poisoned and the clean classifier reduces as the number of training samples increases. This is expected, as the stability of the learning algorithm increases with the number of training data points, limiting the ability of the attacker to perform indiscriminate poisoning attacks. This does not mean that learning algorithms trained with large datasets are not vulnerable to data poisoning, as attackers can still be very successful at performing targeted attacks, focusing on increasing the error on particular instances or creating *backdoors* (Gu et al., 2017). In these scenarios we can also use pGAN to generate more targeted attacks using a surrogate model for the classifier including the subset of samples that the attacker aims to misclassify.

**Multi-class classification:** Finally we performed an error-specific attack on MNIST using the 10 classes. In this case the objective of the attacker is to increase the error of digit 3 being misclassified as a 5. For this, we trained pGAN using a surrogate classifier including only digits 3 and 5, and then, tested against a multi-class classifier trained on $10,000$ data points (see the details of the architecture

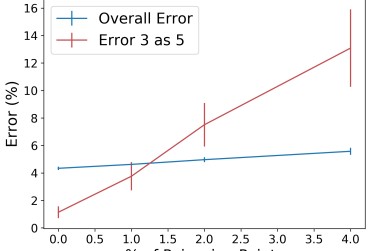 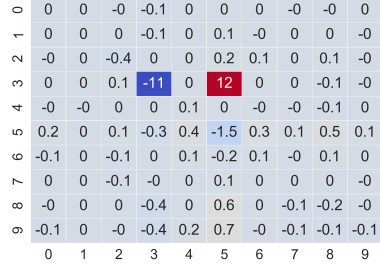

Figure 7: (Left) Overall classification error and error of digit 3 being classified as 5 in MNIST (with all classes), as a function of the attack strength with pGAN. (Right) Difference in the confusion matrix between the clean and the poisoned classifier (4% poisoning).

used in Appendix C). For pGAN we used $\alpha = 0.1$ and $\lambda = 0.9 \cdot \Pr(Y_p)$. We varied the fraction of poisoning points exploring values in the range $0-4\%$. The results in Fig. 7 (left) show that, although the overall test classification error only increases slightly, the test error of digit 3 being classified as a 5 is significantly affected by the attack, increasing from $1.1\%$, when there is no attack, to $13.1\%$ with just $4\%$ of poisoning points. In Fig. 7 (right) we show the average difference in the confusion matrix evaluated on the clean dataset and the poisoned dataset (4% poisoning). We can observe that the detection rate of digit 3 decreases up to 11%, and that this decrease is due to an increase of 12% on the error of digit 3 being incorrectly classified as a 5. This experiment support the usefulness of pGAN to generate targeted attacks, showing that even with a small fraction of poisoning points we can craft successful targeted (error-specific) attacks.

## 4.3 BYPASSING DEFENCES

Apart from the outlier detection defence described previously, we also tested pGAN against 3 different types of defences: First, we considered Sever Diakonikolas et al. (2019), a meta-algorithm for robust optimization that aims to remove outliers and points that can have a negative impact on the learning algorithm at training time. We followed the settings described in Diakonikolas et al. (2019) applying the algorithm separately for each class. We set the value of $\epsilon$, the parameter that controls the fraction of points to be removed, to $0.1$ (results for different values of $\epsilon$ are also shown in Appendix E.1). For the sake of computational tractability we applied Sever using the parameters in the last layer of the target classifier, as these parameters are more influenced by the poisoning attack.[2] Secondly, we considered the defence introduced by (Rubinstein et al., 2009) which relies on Principal Component Analysis (PCA) to detect poisoning points. This model assumes that the clean data lies in a low-rank subspace and that poisoning points will have a large component out of this subspace. As in Sever, this defence has a parameter $\epsilon$ that controls the fraction of points to be rejected. In our experiment we set $\epsilon = 0.05$. Further results for different values of $\epsilon$ are shown in Appendix E.2. Finally we also tested pGAN attack against the label sanitization technique introduced by (Paudice et al., 2018b), which relabels training data points according to a KNN-based algorithm so that a point is relabelled if, at least, $n_k$ neighbours have the same label, different from the label of the evaluated point. Following similar settings as in Paudice et al. (2018b), we set $k = 3$ for KNN and $n_k = 2$.

Using the same settings as in our previous experiments we tested the 3 defences against pGAN attacks with $\alpha = 0.1$ for MNIST and FMNIST. This value of $\alpha$ produced the most effective attacks against the outlier detection defences as shown in Fig. 2. The results in Fig. 8 show that pGAN successfully bypasses all the defences in the two datasets. We can observe that Sever performs worse than the outlier detector defence for MNIST[3] although the degradation with the increasing number of poisoning points follows a similar trend. In the case of FMNIST, Sever is more effective compared to the other defences when the number of poisoning points is reduced, but in this case, the

---

[2]The experimental evaluation included in Diakonikolas et al. (2019) only considers linear classifiers for testing the defence.

[3]Note that the outlier detection defence assumes a stronger model for the defender, which is in control of a fraction of trusted (clean) data points to train the outlier detector.

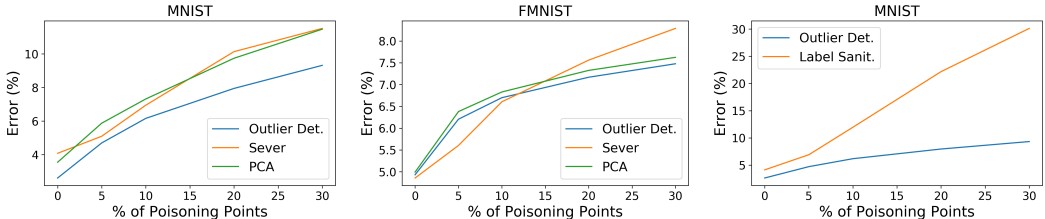

Figure 8: Test classification error (%) as a function of the percentage of poisoning points using pGAN with $\alpha = 0.1$ for MNIST (left) and FMNIST (centre) using outlier detection,Sever (with $\epsilon = 0.1$), and the PCA-based defence (with $\epsilon = 0.05$). Right-most plot compares the test error of outlier detection and label sanitization defences in MNIST.

algorithm degrades faster when we increase the fraction of malicious points. In Appendix E.1 we show that pGAN produces similar effect on Sever for different values of $\epsilon$.

The PCA-based defence performs worse than the outlier detector in MNIST, whereas in FMNIST the difference between the two defences is very small. In both cases, both algorithms degrade in the same way as we increase the number of poisoning points. In Appendix E.2 we show that for larger values of $\epsilon$ this defence performs worse. Finally, in Fig. 8(right) we can observe that label sanitization completely fails to defend against our attack in MNIST. As pGAN produces poisoning points that are correlated, the KNN-based algorithm proposed to do the relabelling is not capable of detecting the poisoning points. Furthermore, some of the genuine points from the target class are incorrectly relabelled, making the problem even worse. Similar results are obtained for FMNIST, as shown in Appendix E.3.

The results in Fig. 8 along with the previous results in Figs. 2-4, show that pGAN not only can generate successful poisoning points at scale even when using detectability constraints but also that pGAN is successful against state-of-the-art defences published in the literature.

## 5 CONCLUSION

The pGAN approach we introduce in this paper allows to naturally model attackers with different levels of aggressiveness and the effect of different detectability constraints on the robustness of the algorithms. This allows to a) study the characteristics of the attacks and identify regions of the data distributions where poisoning points are more influential, yet more difficult to detect, b) systematically generate in an efficient and scalable way attacks that correspond to different types of threats and c) study the effect of mitigation measures such as improving detectability. In addition to studying the tradeoffs involved in the adversarial model, pGAN also allows to naturally study the tradeoffs between performance and robustness of the system as the fraction of poisoning points increases. Our experimental evaluation shows that pGAN effectively bypasses different strategies to mitigate poisoning attacks, including outlier detection, label sanitization, PCA-based defences and Sever algorithm.

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

## A  pGAN Training Algorithm

We train pGAN following a coordinated gradient-based strategy by sequentially updating the parameters of the three components using mini-batch stochastic gradient descent/ascent. The procedure is described in Algorithm 1. For the generator and the discriminator data points are sampled from the conditional distribution on the subset of poisoning labels $\mathbf{Y}_p$. For the classifier, honest data points are sampled from the data distribution including all the classes. We alternate the training for the three components with the $i$, $j$ and $k$ number of steps for the discriminator, classifier, and generator respectively. In practice, we choose $i, j > k$, i.e. we update more often the discriminator and the classifier. For example, in our experiments we set $i, j = 4$ and $k = 1$.

---

**Algorithm 1** pGAN Training

---

**for** number of training iterations **do**
   **for** $i$ steps **do**
      sample mini-batch of $m$ noise samples $\{\mathbf{z}_n|\mathbf{Y}_p\}_{n=1}^m$ from $p_z(\mathbf{z}|\mathbf{Y}_p)$
      get mini-batch of $m$ training samples $\{\mathbf{x}_n\}_{n=1}^m$ from $p_x(\mathbf{x}|\mathbf{Y}_p)$
      update the discriminator by ascending its stochastic gradient

$$\nabla_{\theta_\mathcal{D}} \frac{\alpha}{m} \sum_{n=1}^m \left[\log \mathcal{D}(\mathbf{x}_n|\mathbf{Y}_p) + \log \mathcal{D}(\mathcal{G}(\mathbf{z}_n|\mathbf{Y}_p))\right]$$

   **end for**
   **for** $j$ steps **do**
      sample mini-batch of $m$ noise samples $\{\mathbf{z}_n|\mathbf{Y}_p\}_{n=1}^m$ from $p_z(\mathbf{z}|\mathbf{Y}_p)$
      get mini-batch of $m$ training samples $\{\mathbf{x}_n\}_{n=1}^m$ from $p_x(\mathbf{x})$
      update the classifier by ascending its stochastic gradient

$$\nabla_{\theta_\mathcal{C}} - \frac{1-\alpha}{m} \sum_{n=1}^m \left[\lambda \mathcal{L}_\mathcal{C}(\mathcal{G}(\mathbf{z}_n|\mathbf{Y}_p)) + (1-\lambda)\mathcal{L}_\mathcal{C}(\mathbf{x}_n)\right]$$

   **end for**
   **for** $k$ steps **do**
      sample mini-batch of $m$ noise samples $\{\mathbf{z}_n|\mathbf{Y}_p\}_{n=1}^m$ from $p_z(\mathbf{z}|\mathbf{Y}_p)$
      update the generator by descending its stochastic gradient

$$\nabla_{\theta_\mathcal{G}} \frac{1}{m} \sum_{n=1}^m \left[\alpha \log(1 - D(\mathcal{G}(\mathbf{z}_n|\mathbf{Y}_p)) - (1-\alpha)\mathcal{L}_\mathcal{C}(\mathcal{G}(\mathbf{z}_n|\mathbf{Y}_p))\right]$$

   **end for**
**end for**

---

## B  Synthetic Example: Experimental Settings and Effect on the Decision Boundary

For the synthetic experiment shown in the paper we sample our training and test data points from two bivariate Gaussian distributions, $\mathcal{N}(\mu_0, \Sigma_0)$ and $\mathcal{N}(\mu_1, \Sigma_1)$, with parameters:

$$\mu_0 = \begin{bmatrix} 2.5 \\ -1.0 \end{bmatrix}, \Sigma_0 = \begin{bmatrix} 0.8 & 0.7 \\ 0.7 & 2.0 \end{bmatrix}$$

$$\mu_1 = \begin{bmatrix} 0.5 \\ 1.0 \end{bmatrix}, \Sigma_1 = \begin{bmatrix} 1.0 & 0.3 \\ 0.3 & 1.4 \end{bmatrix}$$

We trained pGAN with 500 training data points for each class with $\lambda = 0.8$ and $\alpha \in [0, 0.2, 0.8, 1]$. We set the number of epochs to $3,000$, the batch-size to $500$, and the parameters in Algorithm 1, $i, j, k = 1$. For the generator and the discriminator we used one-hidden-layer neural networks with Leaky ReLU activation functions. For the classifier we used logistic regression with cross-entropy loss function. The details about the architecture of the three components are detailed in Table 1.

Table 1: pGAN architecture for the Synthetic experiment (Notation: SGD stands for Stochastic Gradient Descent)

| Generator | Architecture: DNN ($2 \times 20 \times 2$) 
 Hidden layer act. functions: Leaky ReLU (negative slope = 0.1) 
 Output layer act. functions: Linear 
 Optimizer: Adam (learning rate = $10^{-4}$) |
|---|---|
| Discriminator | Architecture: DNN ($2 \times 250 \times 1$) 
 Hidden layer act. functions: Leaky ReLU (negative slope = 0.1) 
 Output layer act. functions: Sigmoid 
 Optimizer: SGD (learning rate = $10^{-3}$, momentum = 0.9) |
| Classifier | Architecture: Logistic Regression 
 Loss function $\mathcal{L}_C$: Cross-entropy 
 Optimizer: SGD (learning rate = $10^{-3}$, momentum = 0.9) |

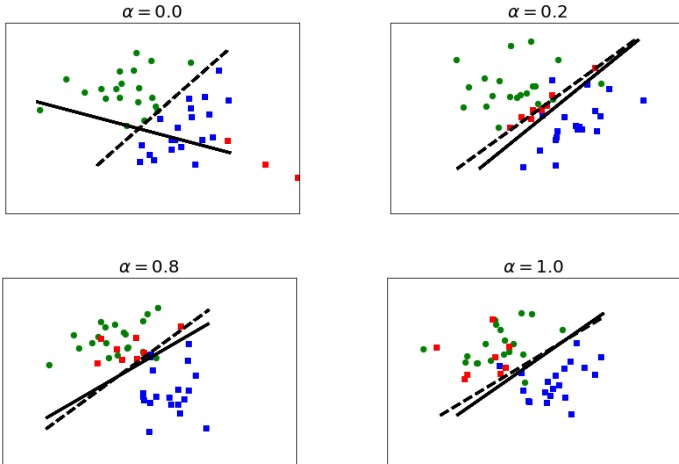

Figure 9: Synthetic experiment: Distribution of genuine (green and blue dots) and poisoning (red dots) data points for different values of $\alpha$. The poisoning points are labelled as green.

In Fig. 9 we show the effect of the poisoning attack on the decision boundary. For testing pGAN we trained a separate logistic regression classifier with 40 genuine training examples (20 per class) and adding extra $20\%$ poisoning points (8 samples). We trained the classifier using Stochastic Gradient Descent (SGD) with a learning rate of $0.01$ for $1,000$ epochs. In this case, no outlier detector is applied to pre-filter the training points. The results in Fig. 9 show that for $\alpha = 0$ the attack is very effective, although the poisoning points depicted in red (which are labelled as green) are far from the genuine distribution of green points. Then, as we increase the value of $\lambda$ the attack is blunt. In this synthetic example, the classifier is quite stable: the number of features is very small (two), and the topology of the problem is simple (the classes are linearly separable and the overlapping between classes is small) and the classifier is simple. Thus, the effect of the poisoning attack when detectability constraints are considered, i.e. $\alpha \neq 0$, is very reduced. Note that the purpose of this synthetic example is just to illustrate the behaviour of pGAN as a function of $\lambda$ rather than showing an scenario where the attack can be very effective.

## C EXPERIMENTAL SETTINGS

Here we provide complete details about the settings for the experiments described in the paper. In Table 2 we show the characteristics of the datasets used in our experimental evaluation. The parameters for training pGAN for MNIST and FMNIST are shown in Table 3, whereas the parameters for CIFAR are described in Table 4. In all cases, for pGAN generator we used (independent) Gaussian noise with zero mean and unit variance.

Table 2: Characteristics of the datasets used in the experiments

| Name | # Training Examples | # Test Examples | # Features |
|---|---|---|---|
| MNIST (3 vs 5) | $6,131/5,421$ | $1,010/892$ | $784$ |
| MNIST (all) | $10,000$ | $10,000$ | $784$ |
| FMNIST (*sneaker* vs *ankle boot*) | $6,000/6,000$ | $1,000/1,000$ | $784$ |
| CIFAR (*automobile* vs *truck*) | $5,000/5,000$ | $1,000/1,000$ | $3,072$ |

Table 3: pGAN architecture for MNIST and FMNIST

| | |
|---|---|
| **Generator** | Architecture: DNN ($100 \times 784 \times 1,024 \times 784$) 
 Hidden layer act. functions: Leaky ReLU (negative slope = 0.1) 
 Output layer act. functions: Tanh 
 Optimizer: Adam (learning rate = $10^{-4}$) 
 Dropout: $p = 0.5$ |
| **Discriminator** | Architecture: DNN ($784 \times 1,024 \times 512 \times 1$) 
 Hidden layer act. functions: Leaky ReLU (negative slope = 0.1) 
 Output layer act. functions: Sigmoid 
 Optimizer: SGD (learning rate = $10^{-3}$, momentum = 0.9) 
 Dropout: $p = 0.5$ |
| **Classifier** | Architecture: DNN ($784 \times 1,024 \times 512 \times 1$) 
 Loss function $\mathcal{L}_{\mathcal{C}}$: Cross-entropy 
 Hidden layer act. functions: Leaky ReLU (negative slope = 0.1) 
 Output layer act. functions: Sigmoid 
 Optimizer: SGD (learning rate = $10^{-3}$, momentum = 0.9) 
 Dropout: $p = 0.5$ |

For MNIST we trained pGAN for $2,000$ epochs using a batch-size of 200, setting $i, j = 4$ and $k = 1$ in Alg. 1. For FMNIST we used similar settings but training for $3,000$ epochs. For CIFAR we trained pGAN using a batch-size of 25 for 300 epochs, with $i, j = 4$ and $k = 1$. Finally, the architecture of the Deep Neural Networks (DNNs) and Convolutional Neural Networks (CNNs) trained to test the attacks is described in Tables 5 and 6.

# D    GENERATION OF POISONING SAMPLES WITH PGAN

In Figs. 10 - 12 we show samples generated with pGAN for different values of $\alpha$ in MNIST, FMNIST and CIFAR respectively. The class labels of the poisoning points are 5 and *ankle boot* and *truck* for each of the datasets. In all cases we can observe that for small values of $\alpha$ (but with $\alpha > 0$), the generated examples exhibit characteristics from the two classes involved in the attack, although pGAN tries to preserve features from the (original) poisoning class to evade detection. For values of $\alpha$ close to 1, the samples generated by pGAN are similar to those we can generate with a standard GAN.

# E    EXTENDED COMPARISON OF DEFENCES

## E.1    SENSITIVITY ANALYSIS FOR SEVER

Here we show the sensitivity analysis of Sever performance to pGAN attack w.r.t. its parameter $\epsilon$, which controls the fraction of training points removed. For this, using the same experimental settings as in the paper, we tested the performance of Sever for $\epsilon = [0.05, 0.1, 0.2]$. The results in Fig. 13 show that the performance of the attack is not very sensitive to the value of $\epsilon$ for both MNIST and FMNIST.

Table 4: pGAN architecture for CIFAR

| | |
|---|---|
| **Generator** | Architecture: DCNN:
• Layer 1: 2D transposed convolutional; input channels: 100; output channels: 1024; kernel size: (2×2); stride: 1; padding: 0; no bias terms; batch normalization
• Layer 2: 2D transposed convolutional; input channels: 1024; output channels: 256; kernel size: (4×4); stride: 2; padding: 1; no bias terms; batch normalization
• Layer 3: 2D transposed convolutional; input channels: 256; output channels: 128; kernel size: (4×4); stride: 2; padding: 1; no bias terms; batch normalization
• Layer 4: 2D transposed convolutional; input channels: 128; output channels: 64; kernel size: (4×4); stride: 2; padding: 1; no bias terms; batch normalization
• Layer 5: 2D transposed convolutional; input channels: 64; output channels: 3; kernel size: (4×4); stride: 2; padding: 1
Hidden layer act. functions: ReLU
Output layer act. functions: Tanh
Optimizer: Adam (learning rate = $10^{-3}$) |
| **Discriminator** | Architecture: DCNN:
• Layer 1: 2D convolutional; input channels: 3; output channels: 64; kernel size: (4×4); stride: 2; padding: 1
• Layer 2: 2D convolutional; input channels: 64; output channels: 128; kernel size: (4×4); stride: 2; padding: 1; no bias terms; batch normalization
• Layer 3: 2D convolutional; input channels: 128; output channels: 256; kernel size: (4×4); stride: 2; padding: 1; no bias terms; batch normalization
• Layer 4: 2D convolutional; input channels: 256; output channels: 512; kernel size: (4×4); stride: 2; padding: 1; no bias terms; batch normalization
• Layer 5: 2D convolutional; input channels: 512; output channels: 1; kernel size: (2×2); stride: 1; padding: 0
Hidden layer act. functions: Leaky ReLU (negative slope = 0.2)
Output layer act. functions: Sigmoid
Optimizer: SGD (learning rate = $1.5 \cdot 10^{-4}$, momentum = 0.5) |
| **Classifier** | Architecture: DCNN:
• Layer 1: 2D convolutional; input channels: 3; output channels: 32; kernel size: (4×4); stride: 2; padding: 1
• Layer 2: 2D convolutional; input channels: 32; output channels: 128; kernel size: (4×4); stride: 2; padding: 1; no bias terms; batch normalization
• Layer 3: 2D convolutional; input channels: 128; output channels: 256; kernel size: (4×4); stride: 2; padding: 1; no bias terms; batch normalization
• Layer 4: 2D convolutional; input channels: 256; output channels: 512; kernel size: (4×4); stride: 2; padding: 1; no bias terms; batch normalization
• Layer 5: 2D convolutional; input channels: 512; output channels: 1; kernel size: (2×2); stride: 1; padding: 0
Hidden layer act. functions: Leaky ReLU (negative slope = 0.2)
Output layer act. functions: Sigmoid
Loss function $\mathcal{L}_{\mathcal{C}}$: Cross-entropy
Optimizer: SGD (learning rate = $10^{-4}$, momentum = 0.5) |

Table 5: Architecture of the classifiers to test the attacks on MNIST, FMNIST and CIFAR.

**Classifier binary MNIST and FMNIST**

Architecture: DNN ($784 \times 1,024 \times 512 \times 1$)
Loss function $\mathcal{L}_\mathcal{C}$: Cross-entropy
Hidden layer act. functions: Leaky ReLU (negative slope = 0.1)
Output layer act. functions: Sigmoid
Optimizer: SGD (learning rate = $10^{-3}$, momentum = 0.9)
Batch size: 200
Epochs: $2,000$
Dropout: $p = 0.5$

**Classifier binary CIFAR**
Architecture: Deep CNN:
  - Layer 1: 2D convolutional; input channels: 3; output channels: 32; kernel size: (3×3); stride: 1; padding: 0; no bias terms; batch normalization; 2D max pooling ($2 \times 2$); dropout: $p = 0.3$
  - Layer 2: 2D convolutional; input channels: 32; output channels: 64; kernel size: (3×3); stride: 1; padding: 0; no bias terms; batch normalization; 2D max pooling ($2 \times 2$); dropout: $p = 0.4$
  - Layer 3: 2D convolutional; input channels: 64; output channels: 128; kernel size: (3×3); stride: 1; padding: 0; no bias terms; batch normalization; 2D max pooling ($2 \times 2$); dropout: $p = 0.5$
  - Layer 4: flattening + fully connected ($512 \times 1$)
Hidden layer act. functions: ReLU
Output layer act. functions: Sigmoid
Loss function $\mathcal{L}_\mathcal{C}$: Cross-entropy
Optimizer: SGD (learning rate = $10^{-3}$, momentum = 0.9)
Batch size: 100
Epochs: $1,000$

Table 6: Architecture of the classifiers to test the attacks on multi-class MNIST (i.e. using all the 10 class labels).

**Classifier multi-class MNIST**

Architecture: DNN ($784 \times 1,024 \times 512 \times 10$)
Loss function $\mathcal{L}_\mathcal{C}$: Cross-entropy
Hidden layer act. functions: Leaky ReLU (negative slope = 0.1)
Output layer act. functions: Softmax
Optimizer: SGD (learning rate = 0.01, momentum = 0.9)
Batch size: 500
Epochs: $1,000$
Dropout: $p = 0.5$

$\alpha = 0.0$ $\alpha = 0.3$

$\alpha = 0.5$ $\alpha = 0.9$

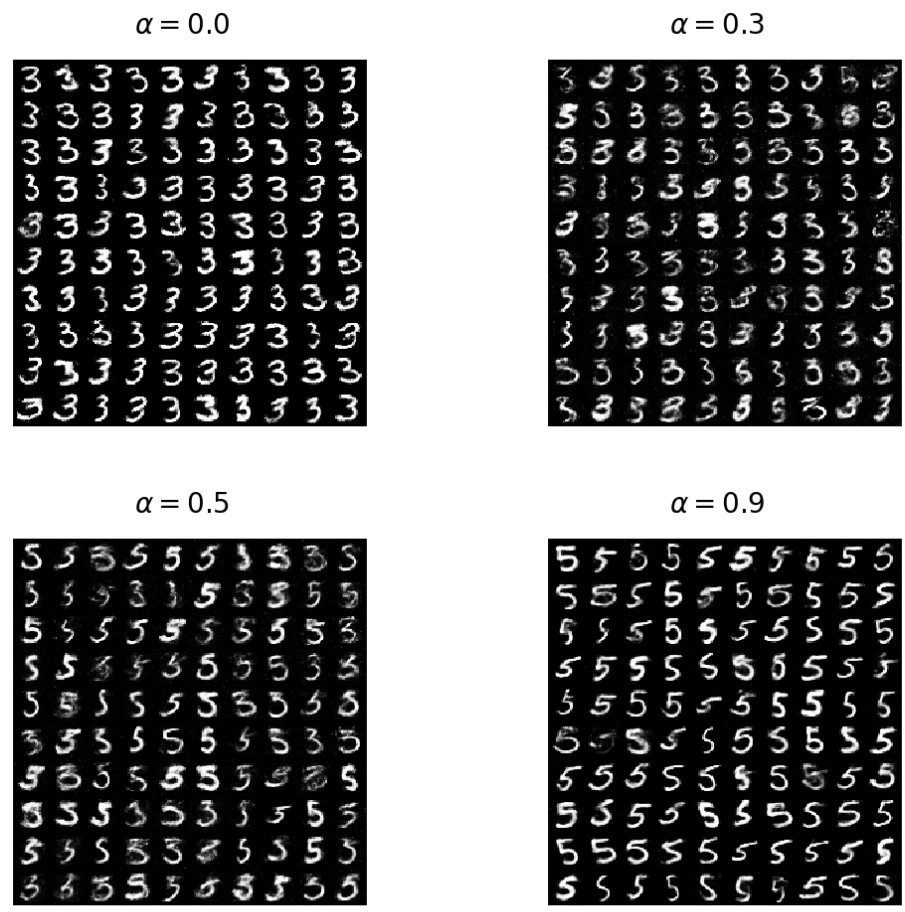

Figure 10: Examples from pGAN on MNIST dataset for different values of $\alpha$.

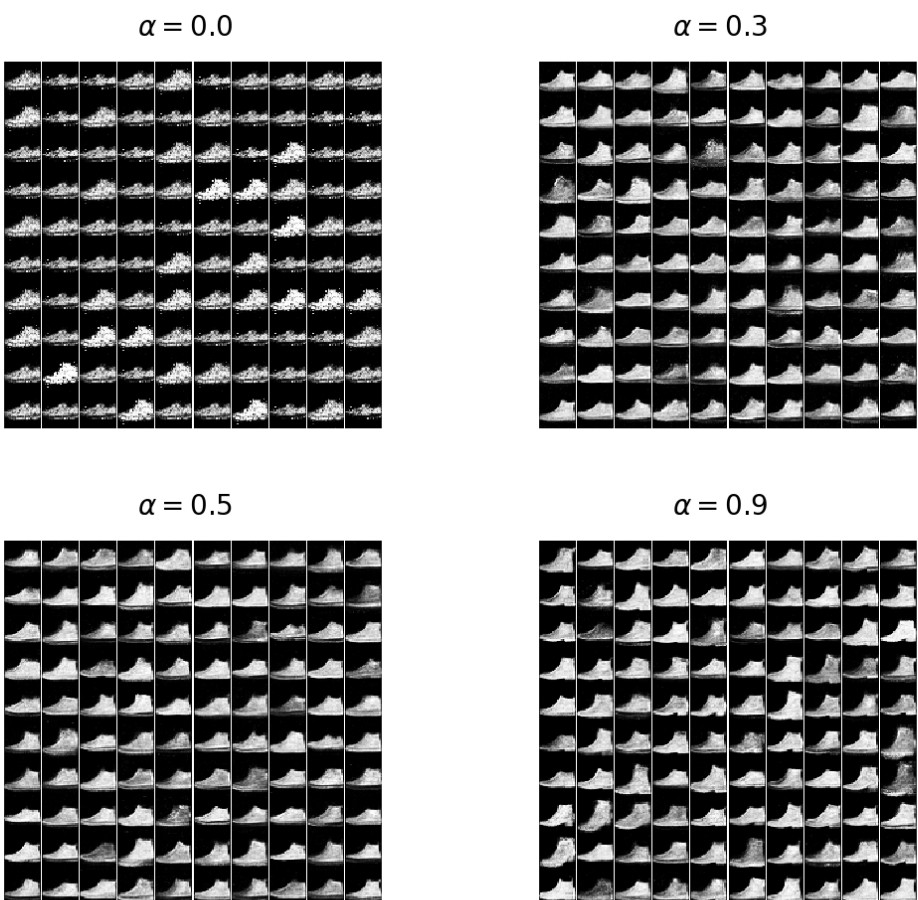

Figure 11: Examples from pGAN on FMNIST dataset for different values of $\alpha$.

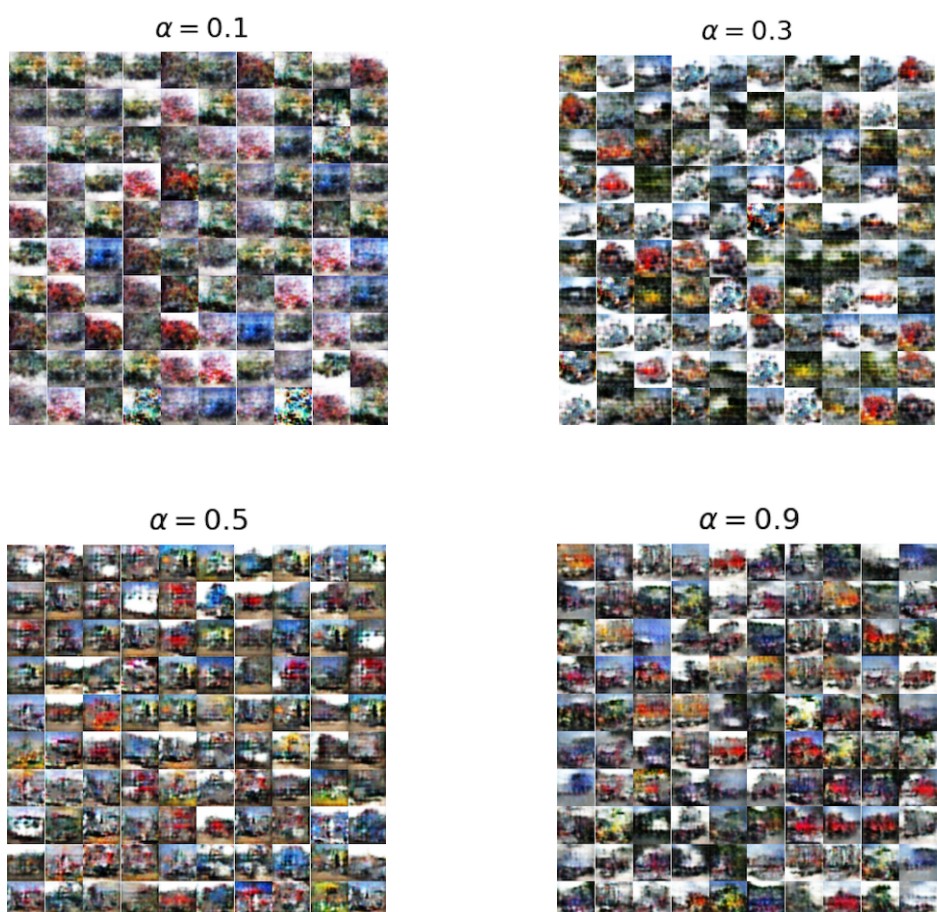

Figure 12: Examples from pGAN with CIFAR dataset for different values of $\alpha$.

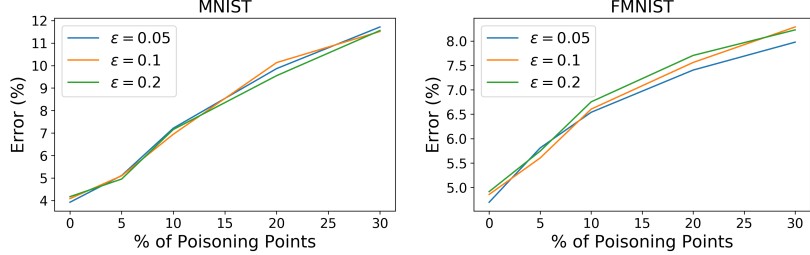

Figure 13: Test classification error(%) for Sever (with $\epsilon = [0.05, 0.1, 0.2]$) as a function of the percentage of poisoning points using pGAN with $\alpha = 0.1$ for MNIST (left) and FMNIST (right).

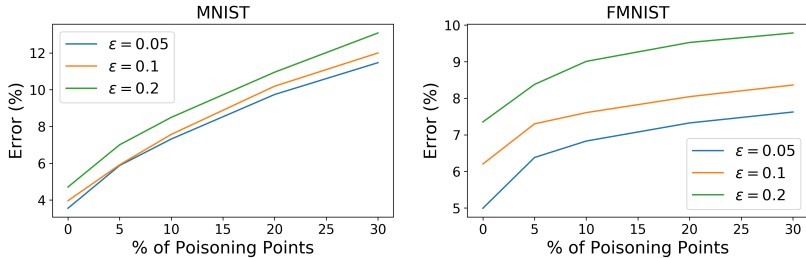

Figure 14: Test classification error(%) for the PCA-based defence (with $\epsilon = [0.05, 0.1, 0.2]$) as a function of the percentage of poisoning points using pGAN with $\alpha = 0.1$ for MNIST (left) and FMNIST (right).

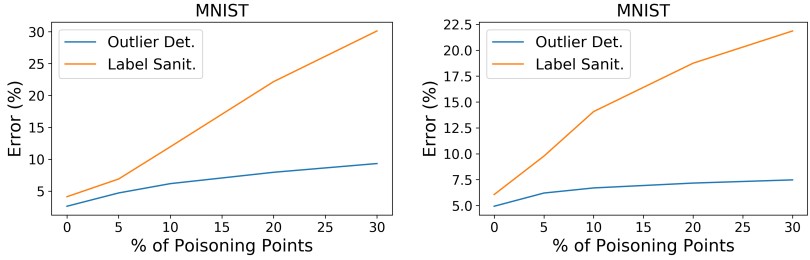

Figure 15: Test classification error(%) for label sanitization and outlier detection defences as a function of the percentage of poisoning points using pGAN with $\alpha = 0.1$ for MNIST (left) and FMNIST (right).

### E.2 SENSITIVITY ANALYSIS FOR THE PCA-BASED DEFENCE

As for Sever, we tested the performance of the PCA defence for different values of $\epsilon$, which as in the previous case, controls the fraction of training points removed. We explored the values $\epsilon = [0.05, 0.1, 0.2]$. The results in Fig. 14 show that, in this case, the performance is further degraded as we increase the value of $\epsilon$, especially for FMNIST.

### E.3 LABEL SANITIZATION

In Fig. 15 we compared the performance of the label sanitization defence and the outlier detection defence for both MNIST and FMNIST. We can observe that label sanitization performs very poorly against pGAN attacks compared to outlier detection. The average test error increases up to 30% with 30% of poisoning points for MNIST, whereas for FMNIST the error increases up to 22% for the same fraction of poisoning points.

