# OpenReview forum: "Poisoning Attacks with Generative Adversarial Nets"
_ICLR.cc/2020/Conference — Reject_

### Official Review · AnonReviewer2 · 2019-10-23
**Official Blind Review #2**

**Rating:** 6

**Review:**


This paper proposed a method pGAN based on Generative Adversarial Networks to generate poisoning examples in order to degrade the performance of classifiers when trained on the poisoned training data. The authors evaluated pGAN on both synthetic datasets and commonly used MNIST and Fashion MNIST datasets in machine learning.

The paper is self-contained and easy to read. My main concern is on the experiment results. The detailed questions are as follows:

Q1: Has the authors tried more complicated datasets such as CIFAR-10 to evaluate the pGAN method? It would make the paper more convincing to add results on more complex datasets.

Q2: Can the authors structure the experimental results with different sections? Currently it is just a single section which is difficult to read.

Q3: The authors noticed that “But, as we decrease the value of α, the distribution of red points shifts towards the region where both green and blue distributions overlap”. This observation is interesting as it finds that the poisoned input tends to lie on the overlap of two classes. But this can easily lead to a defense method: remove those training examples that are close to the other class. This defense mechanism can be used together with other sanitization approaches. So I would like to see how would pGAN perform in this case?

Q4: The authors mentioned “Comparison with existing poisoning attacks in the research literature is challenging: Optimal poisoning attacks as in Munoz-Gonzalez et al. (2017) are computationally very expensive for the size of the networks and datasets used in our experiments in Fig. 2.”.
However, I can not agree because you can simply generate poisoned data and train the neural networks on the poisoned data regardless of the underlying approach that is targeted in generating the poisoned data. This would be an effective baseline to compare. (Correct me if I am wrong here.)

I will change my score if the authors can address my concerns here.

================================================================
Thanks for the rebuttal. I am more convinced now.

**Experience Assessment:**

I have read many papers in this area.

**Review Assessment: Checking Correctness Of Derivations And Theory:**

I carefully checked the derivations and theory.

**Review Assessment: Checking Correctness Of Experiments:**

I carefully checked the experiments.

**Review Assessment: Thoroughness In Paper Reading:**

I read the paper thoroughly.

---

> ### Author Response · Authors · 2019-11-13
> **Updates in the revised version of the paper and reply to the reviewer's comments**
>
> Thank you very much for your feedback. We have revised and updated the paper following your suggestions.
>
> Here are our reply to your comments:
>
> (Q1) Thank you very much for the suggestion. We have included experimental results evaluating pGAN on CIFAR-10. The results are shown in Figure 4 in the updated version of the paper.
>
> (Q2) Following your recommendation, we have structured the section with the experimental results in different subsections.
>
> (Q3) The observation from the reviewer is correct: pGAN aims to inject poisoning points in regions that are close to the decision boundary, especially in those where the data distributions overlap more. Points that are far away from the decision boundary may be detected by the discriminator, outlier detection or other defensive algorithms that could be used.
> Defences aiming to remove points that are close to the decision boundary could be effective to remove poisoning points generated with pGAN, but these defences will suffer from a significant loss in performance, especially when the algorithm is not under attack.
> For example, in SVMs, support vectors are points that are close to the decision boundary. If we remove these points, as suggested by the reviewer, we would obtain different support vectors that would lead to suboptimal solutions with significantly degraded performance.
> With regards to addressing the more general point of the performance of pGAN when state-of-the-art defences are used, we have updated the paper to include a new section (4.3) where we show that pGAN is capable of bypassing 4 different defence mechanisms. This supports the effectiveness of our attack.
>
> (Q4) Munoz-Gonzalez et al. (2017) showed an experiment using a Convolutional neural network with 450,000 parameters, trained with 1,000 training points and injecting 10 poisoning points. In our case, for the experiment with MNIST in Figure 2, we used a deep neural network with more than 40,000,000 parameters, 1,000 training points, injecting up to 400 poisoning points. As the reviewer can observe the scale of the experimental evaluation is significantly different. The computational complexity of the attack in Munoz-Gonzalez et al. (2017) makes the experimental evaluation intractable for the settings considered in our experiments.
> On the other side, Paudice et al. (2018a) showed that, in many cases, if we don’t consider appropriate detectability constraints, the attack points generated by optimal attack strategies formulated as bilevel optimization problems can be effectively filtered out with appropriate outlier detection, resulting in blunt attacks. This is not the case for pGAN, which is capable of bypassing different defences, including the outlier detection scheme proposed by Paudice et al. (2018a).
> Although defences based on outlier detection can be bypassed, as shown by Koh et al. (2017) (Stronger poisoning attacks break data sanitization defences), the complexity of the bilevel problem significantly increases compared to Munoz-Gonzalez et al. (2017). Thus, applying the attack strategy proposed by Koh et al. (2017) is also computationally intractable in our experimental settings.
> One of the main advantages of pGAN is the possibility of generating poisoning attacks at scale with detectability constraints capable of targeting large deep networks, where strategies relying on bilevel optimization have a limited applicability.
>
> Please, let us know if there are aspects that remain unclear or that require further clarification.
>
> Thank you very much.

---

### Official Review · AnonReviewer1 · 2019-10-24
**Official Blind Review #1**

**Rating:** 6

**Review:**

This paper introduces a new generative poisoning attack method against machine learning classifiers. The authors propose pGAN with three components to maximum the error of classification and guarantee undistinguished poisoning data for the discriminator. The experimental results show that the hyperparameter \alpha significantly affects the poisoning data distribution and pGAN leads to specific error in a classification task.

This paper should be weekly accepted, considering the following aspects.

Positive points: (1) The experiments seem solid. The overall performance with different parameters and the corresponding error type have been evaluated. (2) The error-specific and performance-control characteristics of pGAN seem to be interesting. (3) The paper is well organized.

Negative points: (1) The authors should provide more justification on equation-3. Why do the authors directly average different loss for the discriminator and the classifer? (2) The function of the discriminator is not very clear, especially for the classification error test. Does the discriminator exclude the poisoning data according to certain rule? It would make more sense if the classification error measured from the data the discriminator selects. (3) pGAN can produce error-specific attack without sufficient justifications. Why can pGAN lead to the inclination? Is it possible for pGAN to control the specific error tendency? (4) For the error-specific attack task, it would be better to provide an ablation experiment. For example, authors could implement pGAN by ignoring the detectability of the discriminator (i.e. \alpha=0) or typical pGAN when they compare with the label-flip operation. Please explain which component contribute to the error-specific inclination.


**Experience Assessment:**

I do not know much about this area.

**Review Assessment: Checking Correctness Of Derivations And Theory:**

I assessed the sensibility of the derivations and theory.

**Review Assessment: Checking Correctness Of Experiments:**

I assessed the sensibility of the experiments.

**Review Assessment: Thoroughness In Paper Reading:**

I read the paper at least twice and used my best judgement in assessing the paper.

---

> ### Author Response · Authors · 2019-11-13
> **Clarifications and reply to the reviewer's comments**
>
> Thank you very much for your comments and your feedback. We provide our reply to your questions below:
>
> (1) In equation (3) we are using scalarization, a well-known technique to solve multi-objective optimization problems (see for example Boyd’s book “Convex optimization” Ch. 4). In this case, the maximization problem is a multi-objective optimization problem including both the parameters of the discriminator and of the classifier. The parameter alpha controls the importance/priority of each of the objectives.
> The parameter alpha also allows to control the detectability constraints for the attack, which allows us to test the robustness of learning algorithms and defences in different settings, considering more or less aggressive adversaries. This is common in most security settings to test system’s robustness and resilience in different attack scenarios.
>
> (2) In pGAN the discriminator allows to model detectability constraints for the poisoning points. In other words, to evade detection or removal of points by algorithms that defend against poisoning attacks, such as the defences we used in our experiment, we want our attack points to be close to the distribution of the genuine data. However, please, note that the discriminator’s loss is decoupled from the classifier’s loss. In contrast, the generator is the element that competes with both the discriminator and the classifier.
> On the other side, the discriminator does not exclude poisoning data or select any data point but helps to guide the generator to craft poisoning points that are difficult to detect. In other words, the discriminator does not filter out the points that are used to train the classifier during the training of pGAN. It is not clear to us what the reviewer refers to when mentioning measuring the classification error from the data the discriminator selects, as the discriminator does not “select” any data point, but just aim to classify genuine from fake data points.  We would be happy to provide further clarifications on this point if needed.
>
> (3)-(4) To some extent pGAN can control the specific errors produced in the system, as shown both in Figures 5 and 6. But the changes produced in the system may also depend on the characteristics of the dataset and the learning algorithms used.
> pGAN produces poisoning attack points that are close to the decision boundary, “pushing the decision boundary away” from the source class (i.e. the same class as the labels of the poisoning points) towards the samples of the target class. Then, we can expect an increase of the false positive rate, which is shown in Figure 6 (centre). At some point, when the fraction of poisoning points increases significantly the decision boundary starts to change in a different (and possibly more abrupt way), so that the false negatives also start to increase. In Figure 6 (right) this happens when the fraction of poisoning points is larger than 25%.
> In contrast, the label flipping attack is less subtle as it does not consider detectability constraints. The attack points are therefore not necessarily close to the decision boundary, and thus, the changes produced in the algorithm are more unpredictable and affect the errors for the two classes.
>
> If there are points that, in your view, require further clarification, please let us know.
>
> Thank you very much.

---

### Official Review · AnonReviewer3 · 2019-10-25
**Official Blind Review #3**

**Rating:** 3

**Review:**

This paper tackles vulnerability to poisoning. An important subtopic of adversarial ML.
The authors propose using a GAN to generate poisoning data points, as an alternative to existing methods.

While most (or all) of the paper is devoted to illustrate the effectiveness of the approach against *non-protected* ML. My only and biggest concern with this paper is that no defense mechanism has been tested against, and there are many in the literature. (see e.g. Diakonikolas et al ICML 2019).

Thus my question for the rebuttal period: How would pGAN perform when defense mechanisms are deployed during the learning phase? (ideally, a thorough experiment illustrating the strength of pGAN against a few defense mechanisms would help re-evaluating the score)

**Experience Assessment:**

I have published in this field for several years.

**Review Assessment: Checking Correctness Of Derivations And Theory:**

N/A

**Review Assessment: Checking Correctness Of Experiments:**

I assessed the sensibility of the experiments.

**Review Assessment: Thoroughness In Paper Reading:**

I read the paper at least twice and used my best judgement in assessing the paper.

---

> ### Author Response · Authors · 2019-11-13
> **Comparison with different defences included in the revised version**
>
> Thank you very much for the feedback. We have updated the paper and included a new section (4.3) showing how pGAN attacks bypass 4 different defence mechanisms, including outlier detection (as in Paudice et al. 2018a), the PCA-based defence in Rubinstein et al. 2009 (Antidote), Sever (Diakonikolas et al ICML 2019), and label sanitization (Paudice et al. 2018b).
>
> From the reviewer’s comments we noticed that, perhaps, the submitted paper, may not have sufficiently clearly explained that the approach is already targeting defences based on outlier detection and in particular that proposed in Paudice et al. 2018a. We already assume that the defender is in control of a fraction of trusted (clean) data points to train the outlier detector, which is a strong assumption in favour of the defender. To make this point clearer, we have also updated Figure 2 in the paper, showing the performance of pGAN for alpha = 0, i.e. when no detectability constraints are considered. In the Figure, we can observe that both for MNIST and FMNIST the outlier detection is capable of detecting many poisoning points and the effect of the attack is reduced compared with the results for alpha = 0.1.
>
> Different outlier-detection-based defences have already been proposed in the literature, such as Steinhardt et al. 2017 (“Certified defenses for data poisoning attacks”), Koh et al. 2018 (“Stronger data poisoning attacks break data sanitization defenses”) or Paudice et al. 2018a, to cite some. In our experiments we chose the scheme proposed by Paudice et al. 2018a, as it assumes a stronger model for the defender (as mentioned before), which, in our opinion helps to validate the effectiveness of pGAN to craft successful poisoning attacks even in cases where the defender is in control of a fraction of trusted (clean) data points.
>
> Label sanitization (as proposed in Paudice et al. 2018b) completely fails to defend against pGAN attack, as shown in Figure 8 (right). As pGAN produces poisoning points that are correlated, the KNN-based algorithm proposed to do the relabelling is not capable of detecting the poisoning points. Moreover, some of the genuine points from the target class are incorrectly relabelled, making the problem even worse.
>
> The PCA-based defence proposed by Rubinstein et al. 2009 (Antidote) is also not capable of mitigating pGAN attack. The detectability constraints included in our model prevents this defence to detect the generated poisoning points. In the supplement we have included an analysis of the sensitivity of this algorithm to the threshold to discard training points. We can observe that the error increases as we increase this threshold.
>
> The “Sever” defence (Diakonikolas et al. 2019 ICML) is also not robust against pGAN attack. In Figure 8 (left) we can observe that the defence performs worse than the outlier detector and that, when the algorithm is not under attack, the performance slightly decreases, as the algorithm is removing genuine data points that are significant for the training process. For FMNIST, Sever outperforms the outlier detector when the number of poisoning points is reduced, although the degradation of the algorithm as we increase the fraction of poisoning points is faster compared to the outlier detector and the PCA-based defence. In the supplement we included the sensitivity analysis w.r.t. the parameter that controls the fraction of points to be discarded. We can observe that, in this case, the difference in performance is not significant for the different values explored for this threshold.
>
> In summary, the revised paper (see the new version uploaded) now provides a comprehensive comparison of different defence mechanisms and shows the effectiveness of pGAN to bypass all of them. First in Figure 2 we show the effect of the attack for different values of alpha tested against the outlier-detection-based defence. Then, we have provided an empirical evaluation of pGAN against 4 different defence mechanisms both in MNIST and FMNIST, showing how our attack bypasses all of these defences.
>
> We thank the reviewer for this valuable comment, which has certainly helped us to improve the paper. We hope that the score can be revised to reflect this improvement.
>
> Thank you very much.

---

### Decision · Program_Chairs · 2019-12-19

**Decision:**

Reject

**Comment:**

This paper proposes a GAN-based approach to producing poisons for neural networks.  While the approach is interesting and appreciated by the reviewers, it is a legitimate and recurring criticism that the method is only demonstrated on very toy problems (MNIST and Fashion MNIST).  During the rebuttal stage, the authors added results on CIFAR, although the results on CIFAR were not convincing enough to change the reviewer scores; the SOTA in GANs is sufficient to generate realistic images of cars and trucks (even at the ImageNet scale), while the demonstrated images are sufficiently far from the natural image distribution on CIFAR-10 that it is not clear whether the method benefits from using a GAN.   It should be noted that a range of poisoning methods exist that can effectively target CIFAR, and SOTA methods (e.g., poison polytope attacks and backdoor attacks) can even target datasets like ImageNet and CelebA.